# A CDC25 family protein phosphatase gates cargo recognition by the Vps26 retromer subunit

Tie-Zhong Cui[1], Tabitha A Peterson[2], Christopher G Burd[1]*

[1]Department of Cell Biology, Yale School of Medicine, New Haven, United States; [2]Department of Molecular Physiology and Biophysics, University of Iowa, Iowa City, United States

**Abstract** We describe a regulatory mechanism that controls the activity of retromer, an evolutionarily conserved sorting device that orchestrates cargo export from the endosome. A spontaneously arising mutation that activates the yeast (*Saccharomyces cerevisiae*) CDC25 family phosphatase, Mih1, results in accelerated turnover of a subset of endocytosed plasma membrane proteins due to deficient sorting into a retromer-mediated recycling pathway. Mih1 directly modulates the phosphorylation state of the Vps26 retromer subunit; mutations engineered to mimic these states modulate the binding affinities of Vps26 for a retromer cargo, resulting in corresponding changes in cargo sorting at the endosome. The results suggest that a phosphorylation-based gating mechanism controls cargo selection by yeast retromer, and they establish a functional precedent for CDC25 protein phosphatases that lies outside of their canonical role in regulating cell cycle progression.

*For correspondence:
christopher.burd@yale.edu

**Competing interests:** The authors declare that no competing interests exist.

## Introduction

Endocytosis mediates internalization of the plasma membrane via vesicles that deliver their content to the endosomal system, a network of related organelles that undergo maturation to generate a terminal endosome that fuses with the lysosome. During endosome maturation, integral membrane proteins are either retained, leading their eventual turnover in the lysosome, or they are exported and delivered to other organelles for re-use. Molecular sorting reactions in the endosome thus play a fundamental role in controlling the composition of the plasma membrane by determining which molecules are recycled back to the plasma membrane and which are degraded.

The retromer protein complex is an evolutionarily conserved sorting device that orchestrates export of integral membrane proteins from the endosome via tubular and vesicular ETCs (*Burd and Cullen, 2014*; *Chi et al., 2015*). Retromer is composed of three proteins: Vps35, which resembles an alpha solenoid similar to well characterized vesicle coat proteins such as clathrin; Vps26 (standard name: Pep8), which binds the N-terminal region of Vps35; and Vps29, which binds the C-terminal region of Vps35 (*Collins et al., 2008*; *Hierro et al., 2007*; *Norwood et al., 2011*). The full range of molecular functions that are executed by each retromer subunit have yet to be definitively established, and this is a major obstacle for understanding the mechanisms that underpin retromer-mediated cargo export from the endosome. The helical repeats of Vps35 are thought to provide binding sites for cargo and accessory factors, such as the Snx3 and Snx27 sorting nexins (*Harrison et al., 2014*; *Hierro et al., 2007*; *Liu et al., 2012*) and, in mammals, the WASH complex, an activator of the Arp2/3 actin polymerization complex (*Harbour et al., 2012*; *Jia et al., 2012*). Vps26 is structurally related to α- and β-arrestins (*Collins et al., 2008*; *Shi et al., 2006*), which function as sorting adapters for G protein-coupled receptors (GPCRs) in clathrin-mediated endocytosis, raising the

possibility that it may function in the selection of integral membrane proteins for export. Given the central role that retromer plays in controlling plasma membrane composition and organelle biogenesis, it is logical to expect that retromer activities would be regulated in order to integrate its functions with cell physiology (*Seaman, 2012*), but examples of such regulation have yet to be identified.

In this study, we used genetic selection in budding yeast (*Saccharomyces cerevisiae*) to identify gene products that control plasma membrane residence of integral membrane proteins. One mutant obtained displays a loss of retromer-dependent plasma membrane recycling of multiple integral plasma membrane proteins. This phenotype results from a gain-of-function mutation in the dual specificity protein phosphatase (*Camps et al., 2000*; *Patterson et al., 2009*), *MIH1*, encoding the *S. cerevisiae* homolog of *CDC25* (*Pal et al., 2008*). We show that Vps26 directly recognizes a retromer sorting signal and that the phosphorylation state of Vps26, controlled by Mih1, modulates the affinity of retromer for the Chs3 recycling signal.

## Results and discussion

### Isolation of yeast mutants deficient in endosomal trafficking

To identify factors that regulate post-Golgi trafficking of integral membrane proteins, we harnessed the activity of yeast chitin synthase 3 (Chs3), an integral membrane enzyme that is trafficked between Golgi and endosomal compartments and the plasma membrane. Delivery of Chs3 to the cell surface is mediated by transport vesicles that are coated with the 'exomer' protein complex (*Wang et al., 2006*). In cells lacking functional exomer, Chs3 is constitutively retained within the cell by a Golgi-endosome trafficking circuit involving a clathrin adapter protein complex 1 (AP1)-dependent retention mechanism (*Valdivia et al., 2002*; *Wang et al., 2006*). In cells that lack both functional exomer and AP1 sorting complexes, Chs3 fails to be retained and it is delivered to the plasma membrane via the constitutive secretory pathway (*Valdivia et al., 2002*) (*Figure 1A*). Whereas growth of exomer-deficient cells is resistant to calcofluor white (CFW), a cytotoxic molecule that binds chitin in the cell wall (*Roncero and Durán, 1985*), growth of cells lacking exomer and AP1 is exquisitely sensitive to CFW in the medium (*Valdivia et al., 2002*) (*Figure 1*).

We used selection on CFW-containing growth medium to obtain CFW-resistant ('CFW^R') mutants and then identified those exhibiting altered distribution of Chs3 (*Figure 1B,C*). To bias the selection away from mutations that affect exomer- and AP1-mediated trafficking, we employed a strain deleted for both an exomer subunit (*CHS6*) and an AP1 subunit (*APL2*), *chs6Δapl2Δ*. As previously reported (*Valdivia et al., 2002*), these cells are sensitive to CFW (*Figure 1C*), however, we observed spontaneously arising CFW^R colonies after several days on solid growth medium containing 100 μg/ml CFW (*Figure 1C*); we considered these to be good candidates for strains carrying spontaneously arising mutations that cause a reduction in cell wall chitin. Indeed, cells of each of these strains exhibited reduced capacity to bind CFW, as determined by visualizing cells by ultra-violet illumination (not shown).

Complementation tests against a panel of mutants implicated in Chs3-dependent chitin deposition (*chs3△*, *chs4△*, *chs5△*, *chs7△*, and *pfa4△*) (*Lam et al., 2006*; *Ono et al., 2000*; *Trilla et al., 1999*; *Wang et al., 2006*) revealed that, as expected, the majority of CFW^R mutants (96%) failed to be complemented (i.e., diploid cells remained CFW^R) by any single 'tester' strain (*Figure 1B*), strongly suggesting that lesions in the 'tester' genes are responsible for the observed CFW resistance. These mutants were discarded, leaving a subset of 30 strains that were not complemented by any of the tester strains. To determine if Chs3 trafficking is perturbed in cells of any of these strains, we compared localization of a Chs3-GFP fusion protein in each mutant strain to that of the original parent strain. In a population of parental *chs6△apl2△* cells, Chs3-GFP is observed at the bud neck, intracellular punctae (Golgi and/or endosome compartments), and there is a faint signal in the vacuole lumen of most cells (*Figure 2A*). In 19 out of 30 CFW^R strains examined, there are fewer and less bright punctae, the intensity of Chs3-GFP fluorescence at the bud neck is reduced, and the GFP signal in the vacuole lumen is substantially increased (an example of one such mutant is shown in *Figure 2A*).

Whole-genome sequencing of the most severe vacuolar localization mutant led to the identification of a cytosine-to-adenine mutation in the *MIH1* ORF that causes a S162R substitution in the Mih1

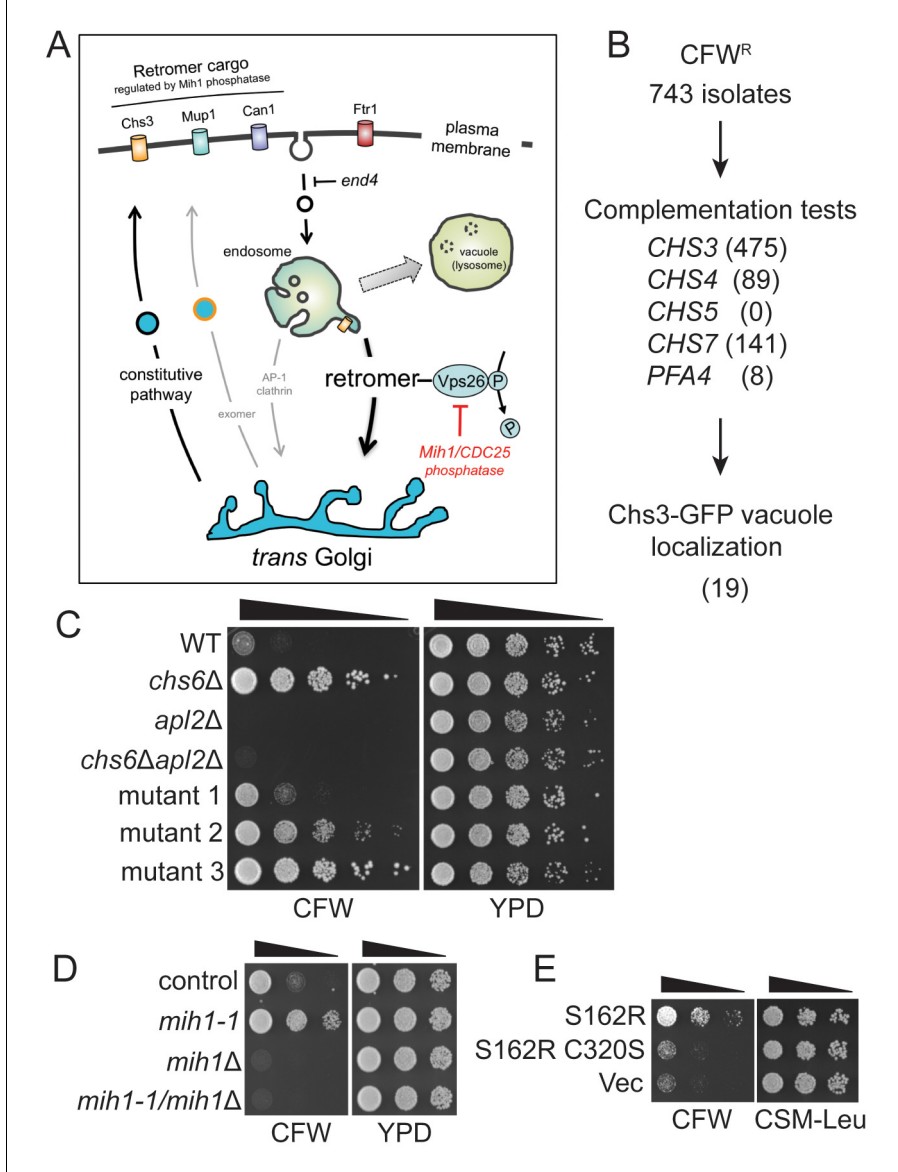

**Figure 1.** Selection of yeast mutants resistant to calcofluor white. (**A**) Schematic diagram of post-Golgi trafficking pathways. In wild-type cells, Chs3 is retained at the Golgi-endosome interface via AP-1 dependent retrieval from the endosome, and is delivered to the plasma membrane via Golgi-derived exomer coated vesicles (gray arrows). In cells used for the genetic selection in this study, chitin synthase 3 (Chs3) is diverted into the constitutive secretory pathway due to deletions of the Chs6 exomer subunit and the Apl2 AP-1 subunit (*chs6Δapl2Δ*). Endocytosis of Chs3 and other plasma membrane proteins, which is attenuated by the *end4* mutation, delivers the internalized proteins to the endosome, where some are sorted by retromer into a recycling pathway, while others are retained in the endosome and delivered to the lysosome-like vacuole and degraded. (**B**) Strategy to identify novel mutations affecting Chs3 trafficking. The flowchart lists the number of mutants remaining after each step of selection. (**C**) Representative calcofluor white (CFW) resistant strains. Serial dilutions (1:10; indicated by the black triangle above the photograph) of the indicated strains were spotted onto YPD medium with or without 100 μg/ml CFW. The plates were incubated for three days at 30°C and then photographed. (**D**) Mutations in *MIH1* alter sensitivity of growth to CFW. Serial dilutions (indicated by a triangle above the photograph) of control cells of the indicated genotypes were spotted onto rich medium with or without 50 μg/ml CFW for three days at 30°C. The genotype of the control strain is: *chs6Δapl2Δ*. The *chs6Δapl2Δmih1-1* strain was constructed by introducing the *mih1-1* lesion de novo in the original *chs6Δapl2Δ* parent strain. The *mih1Δ* designation indicates that the wild-type *MIH1* locus was deleted in the original parent strain (*chs6Δapl2Δmih1Δ*). The *mih1-1/mih1Δ* designation indicates that the *mih1-1* allele was deleted in the original *mih1-1* mutant strain. (**E**) Mih1 phosphatase activity is required for

*Figure 1 continued on next page*

*Figure 1 continued*

CFW resistance conferred by the *mih1-1* mutation. Serial dilutions of *chs6Δapl2Δmih1Δ* cells containing plasmids expressing the *mih1-1* allele ('S162R'), a double mutant protein (S162R C320S) in which a second mutation was introduced that ablates catalytic activity, or empty vector ('Vec'), were spotted onto complete synthetic medium with or without 50 µg/ml CFW for three days at 30°C. The host strain is the *chs6Δapl2Δmih1Δ* strain background.

The following figure supplement is available for figure 1:

**Figure supplement 1.** Regulation of membrane trafficking by Mih1 is not dependent on activated Cdc28.

protein. That this mutation is responsible for the phenotypes that we selected was confirmed by the identical phenotypes of cells in which this mutation was constructed de novo in the parent strain, and by deletion of the *MIH1 locus* in the original mutant strain, which reverted the enhanced vacuolar targeting of Chs3 (*Figure 2*) and conferred enhanced sensitivity to CFW (*Figure 1D*). We termed this allele *mih1-1*.

MIH1 is homologous to *CDC25*, first discovered in *Schizosaccharomyces pombe,* which encodes a dual-specificity protein phosphatase that dephosphorylates the Cdc2 cyclin-dependent kinase (CDK; Cdc28 in *S. cerevisiae*) to activate it (*Pal et al., 2008*; *Patterson et al., 2009*; *Russell and Nurse, 1986*). The substitution caused by the *mih1-1* mutation lies outside of the phosphatase motif, however, the phosphatase activity of Mih1 is necessary for the observed phenotypes of the *mih1-1* mutant, because introduction of a second mutation, cysteine(320) to serine, which changes an active site residue required for catalysis (*Yano et al., 2013*), ablates CFW$^R$ caused by the *mih1-1* mutation (*Figure 1E*). Mih1 undergoes cell cycle regulated phosphorylation at numerous sites and this is correlated with inhibition of its activity toward Cdc28/CDK (*Pal et al., 2008*). While S162 is not a known site of phosphorylation, several residues in the vicinity of S162 are proposed to be phosphorylated by protein kinase C (PKC) (*Yano et al., 2013*), a protein kinase shown to regulate trafficking of Chs3 (*Valdivia and Schekman, 2003*). Thus, we speculate that the S162R mutation may impinge on regulation of Mih1 activity.

As the proposed function of Mih1 is to activate Cdc28/CDK by dephosphorylating an active site tyrosine residue (*Pal et al., 2008*), we tested the possibility that the *mih1-1* associated trafficking phenotypes are due to hyper-activation of Cdc28/CDK. Immunoblotting with anti-phospho-CDK antiserum did not reveal a change in the amount of phosphorylated Cdc28/CDK in lysates from cultures of *chs6Δapl2Δ* versus *chs6Δapl2Δmih1-1* cells (*Figure 1—figure supplement 1*), probably because multiple phosphatases act redundantly to activate Cdc28/CDK (*Kennedy et al., 2016*). Accordingly, this hypothesis was further tested by deleting the gene encoding the Swe1 kinase that phosphorylates Cdc28/CDK to inhibit its activity (*Lianga et al., 2013*; *Pal et al., 2008*); if the *mih1-1* allele exerts its effect via activated Cdc28/CDK, deletion of *SWE1* should revert the CFW$^R$ and Chs3-GFP trafficking phenotypes. However, we observed that CFW resistance and Chs3-GFP localization are unaffected by the *swe1Δ* mutation (*Figure 1—figure supplement 1*). In addition, deletion of *SWE1* in the original *chs6Δapl2Δmih1-1* mutant strain (*chs6Δapl2Δmih1-1swe1Δ*) has no effect on CFW sensitivity or Chs3-GFP localization (*Figure 1—figure supplement 1*). These data indicate that the consequences of the *mih1-1* mutation on Chs3 trafficking are not exerted via activated Cdc28/CDK.

## Mih1 regulates recycling of plasma membrane proteins

The data suggest that CFW resistance is caused by an increased rate of turnover of Chs3 in the vacuole of *chs6Δapl2Δmih1-1* cells. Anti-GFP immunoblotting of cell extracts confirmed this; in lysates of *chs6Δapl2Δmih1-1* cells expressing Chs3-GFP, GFP is cleaved from ~62% of the Chs3-GFP fusion protein, but just ~37% is processed to GFP in lysates from the *chs6Δapl2Δ* parent (*Figure 2*). Importantly, in parental cells lacking *MIH1* (*chs6Δapl2Δmih1Δ*), GFP is cleaved from just ~22% of the Chs3-GFP fusion protein with a concomitant accumulation of full length Chs3-GFP relative to parental cells (*Figure 2*). Thus, the *mih1-1* and *mih1Δ* alleles have opposing consequences, indicating that native Mih1 is a physiological regulator of Chs3 trafficking.

In order to determine if the *mih1-1* mutation uniquely affects trafficking of Chs3, we examined localization and processing of GFP-tagged forms of several nutrient transporters that are maintained

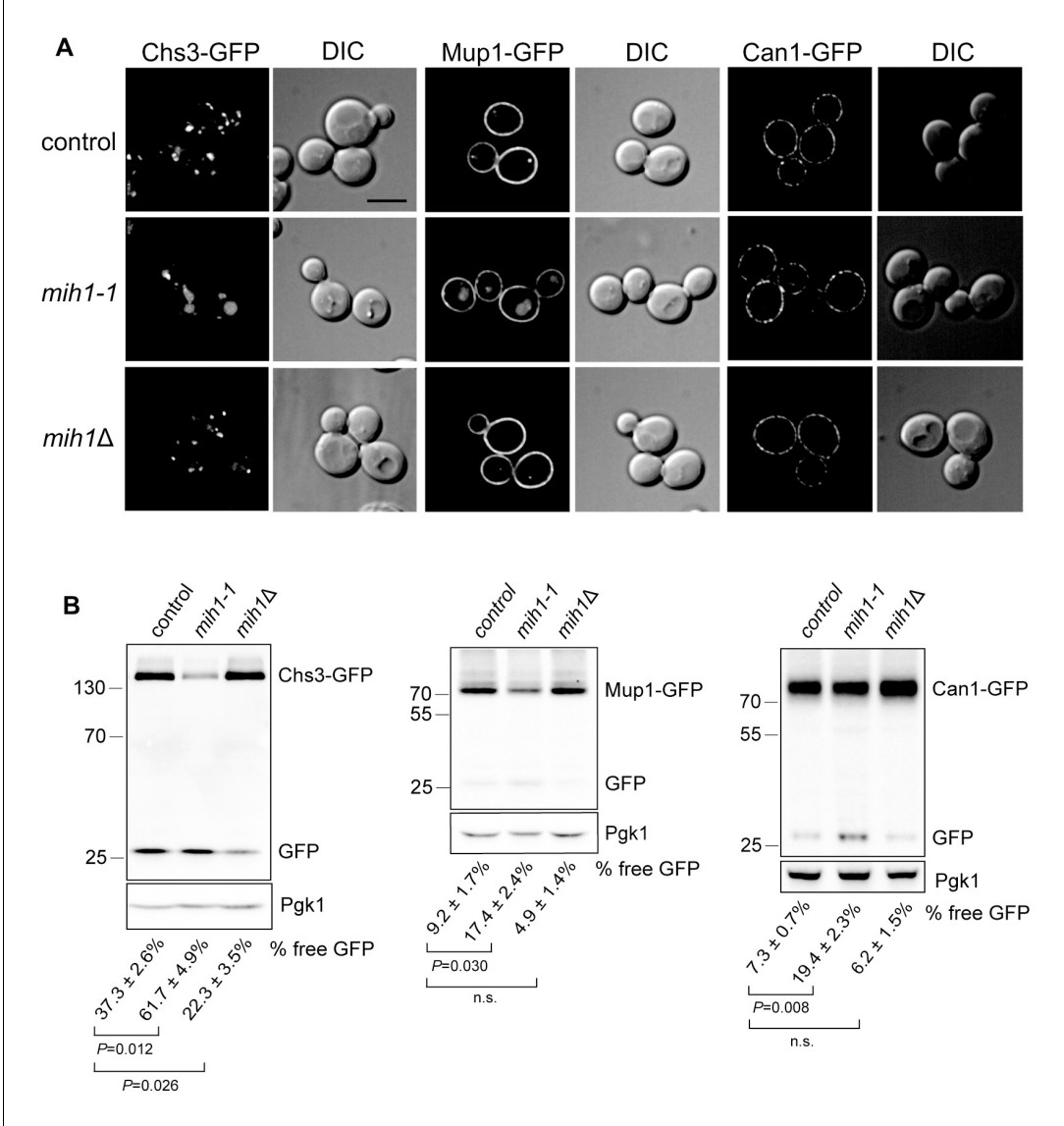

**Figure 2.** Mih1 regulates turnover of a subset of plasma membrane proteins. (**A**) Localization of Chs3-GFP, Mup1-GFP, and Can1-GFP in control cells (*chs6Δapl2Δ*), *mih1-1* (*chs6Δapl2Δmih1-1*) and *mih1Δ* (*chs6Δapl2Δmih1Δ*). A representative medial optical plane of a z-series is shown. Scale bar, 5 µm. (**B**) Increased proteolysis of Chs3-GFP, Mup1-GFP, and Can1-GFP in *mih1-1* (*chs6Δapl2Δmih1-1*) cells and decreased proteolysis in *mih1Δ* (*chs6Δapl2Δmih1Δ*) cells. Mup1-GFP and Can1-GFP expressing cells were grown in methionine- or arginine-deficient medium, respectively, to promote plasma membrane recycling. A representative anti-GFP immunoblot of cell lysates of the indicated strains is shown. The number below each lane indicates the mean proportion (+s.e.m.) of free GFP to the total GFP signal in the lane calculated from a minimum of three independent measurements. Anti 3-phosphoglycerate kinase (Pgk1) blotting was used to control for loading. Statistical significance for pairwise comparisons is indicated; n.s., not statistically significant. The positions of molecular weight (kDa) protein standards is indicated on the left side of the gel.

The following figure supplement is available for figure 2:

**Figure supplement 1.** Localization and steady state levels of Ftr1-GFP, Vps10-GFP and GFP-Snc1 in control (*chs6Δapl2Δ*), *mih1-1* (*chs6Δapl2Δmih1-1*), and *mih1Δ* (*chs6Δapl2Δmihμ*) cells.

at the plasma membrane via endocytic recycling in *chs6Δapl2Δmih1-1* and *chs6Δapl2Δmih1Δ* cells. As observed for Chs3-GFP in the original *mih1-1* mutant, increased vacuole localization and proteolytic processing of Mup1-GFP (a GFP tagged methionine transporter) and Can1-GFP (a GFP tagged arginine transporter) is observed (*Figure 2*). In addition, deletion of *MIH1* (i.e., *chs6Δapl2Δmih1Δ* cells) results in an increase in the levels of full-length Mup1-GFP and Can1-GFP (*Figure 2*), while the localization, levels, and proteolytic processing of Ftr1-GFP (a GFP tagged iron transporter), Vps10-GFP (a GFP-tagged integral membrane protein that is trafficked from the endosome to the Golgi by retromer), and GFP-Snc1 (a GFP tagged v-SNARE of secretory vesicles) are not affected by mutations in *MIH1* (*Figure 2—figure supplement 1*). These results indicate that Mih1 regulates recycling of a subset of proteins that transit the plasma membrane.

In principle, the *mih1-1* mutation could affect sorting upon export from the Golgi or sorting at the endosome, after cargo has been internalized by endocytosis. To distinguish between these modes, we monitored localization and processing of Chs3-GFP and Mup1-GFP in a strain in which the *end4-1* mutation was introduced into the *chs6Δapl2Δmih1-1* strain (*chs6Δapl2Δmih1-1 end4-1*). This mutation substantially limits endocytosis via the clathrin-dependent pathway (*Munn and Riezman, 1994*), resulting in the accumulation of recycling proteins on the plasma membrane (*Strochlic et al., 2007*). In cells carrying the *end4-1* mutation, the Chs3-GFP and Mup1-GFP (*Figure 3A,C*) fluorescence signals in the vacuole is decreased, and the proportion of GFP cleaved from the fusion proteins also decreases (*Figure 3B,D*). These data indicate that Mih1 acts downstream of endocytosis to regulate the abundance of Chs3 and Mup1 on the plasma membrane, strongly suggestive of a role in endocytic recycling.

## Plasma membrane recycling of Chs3 requires retromer

Sorting nexins are components of endosomal recycling/retrograde sorting devices (*Cullen and Korswagen, 2012*) and we therefore speculated that one or more sorting nexins are required for Chs3 recycling in the absence of AP1-mediated recycling (i.e., as in *chs6Δapl2Δ* parental cells). To test this, CFW resistance of cells deleted of a sorting nexin with an established role in endocytic recycling introduced into the parental *chs6Δapl2Δ* Chs3-GFP was examined (*Figure 4A*). This screen revealed that deletion of *vps5Δ* and *vps17Δ*, but no other sorting nexin deletion, including *snx3Δ*, which has been shown to ablate plasma membrane recycling of Ftr1-GFP (*Strochlic et al., 2007*), conferred resistance to CFW comparable to the *chs6Δapl2Δmih1-1* strain. It is noteworthy that Vps5 and Vps17 form a SNX-BAR heterodimer that functions as a component of the retromer sorting complex (*Horazdovsky et al., 1997*; *Seaman et al., 1998*) and, indeed, deletion of the gene encoding each retromer subunit also confers resistance to CFW (*Figure 4A*) and Chs3-GFP is localized prominently to the vacuole lumen (*Figure 4B*). These results indicate that Chs3 is exported from the endosome via a retromer-dependent recycling pathway.

## Regulation of retromer by Mih1

Deficient trafficking of multiple proteins (Chs3, Mup1, and Can1) in the endosomal system of *mih1-1*, *mih1Δ*, and retromer null cells leads us to speculate that Mih1 regulates the activity of a general endosomal sorting and trafficking device – retromer – via dephosphorylation of one or more retromer subunits. To test this, we immunopurified GFP-tagged forms of each retromer subunit from *chs6△apl2△* and *chs6△apl2△mih1△* cells, and then probed the precipitates with a pan phosphoryl–amino acid antiserum (*Figure 4C*). We observe an accumulation of phosphorylated Vps26-GFP in cells lacking Mih1 (*chs6△apl2△mih1△*); we also note that a phosphorylated form of Vps35-GFP was detected, but only in a subset of these experiments (not shown). Phosphorylated Vps26-GFP could also be detected after GFP immuno-purification from wild-type cell lysate, but not in lysate prepared from *mih1-1* cells (*Figure 4D*), consistent with the *mih1-1* allele conferring enhanced activity, as predicted from the genetic analyses.

Changes in the phosphorylation status of Vps26 in *mih1-1* and *mih1Δ* cells raise the possibility that phosphorylated Vps26 is a substrate of Mih1, though this is unexpected, as CDC25 family phosphatases are thought to be dedicated toward CDKs (*Boutros et al., 2007*). We tested this by assaying the activities of purified Mih1, Mih1-1, and the catalytically inactive C320S form of Mih1, toward phosphorylated Vps26. Mih1 proteins were purified from yeast strains that overexpress each protein (*Figure 4E*), and Vps26-GFP was purified from a *mih1Δ* strain. The results clearly show that the wild-

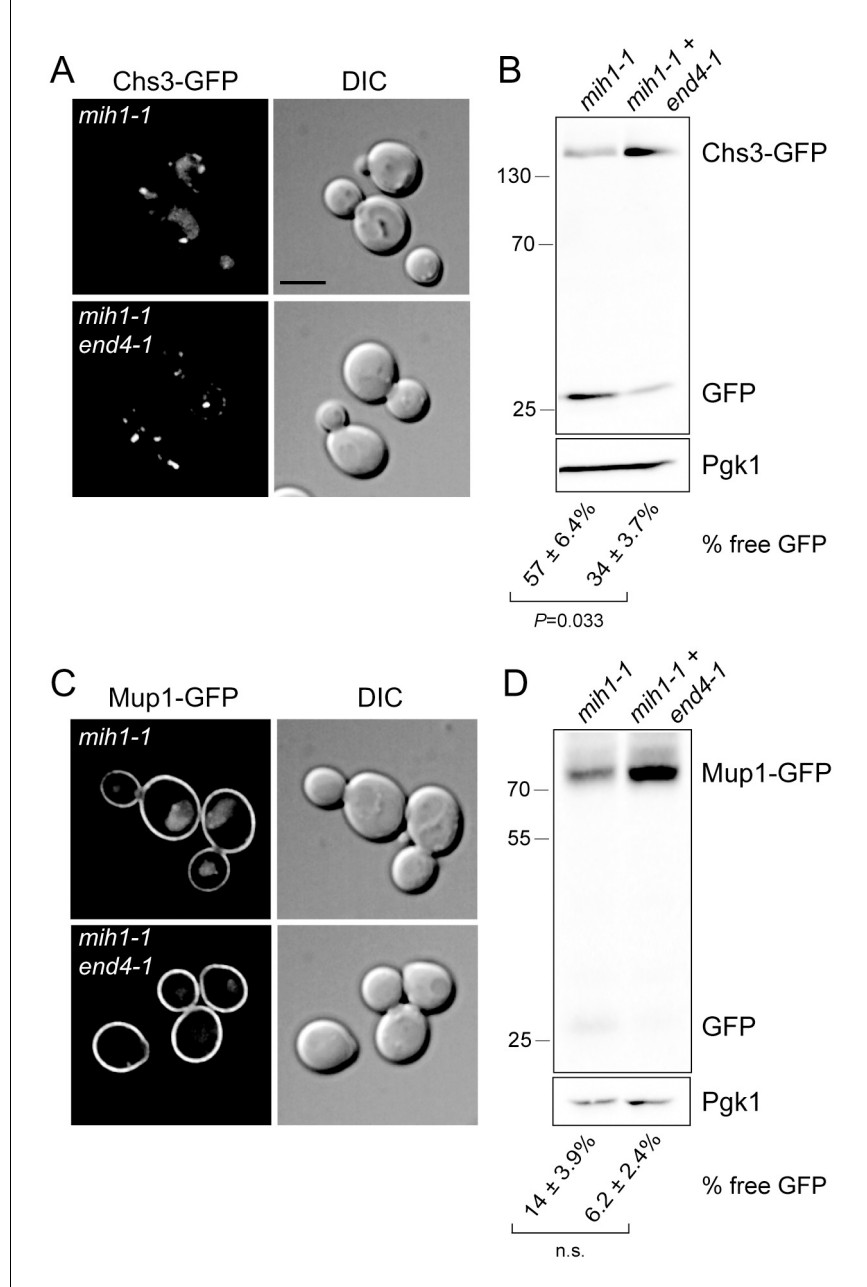

**Figure 3.** Mih1 regulates protein sorting at the endosome. (**A**) Endocytosis promotes vacuolar localization of Chs3-GFP in *mih1-1* cells. Micrographs showing Chs3-GFP in *mih1-1* (*chs6Δapl2Δmih1-1*) and *mih1-1end4-1* (*chs6Δapl2Δmih1-1end4-1*) cells are shown. A representative medial optical plane of a z-series is shown. Scale bar, 5 μm. (**B**) Cleavage of GFP from Chs3-GFP is attenuated by the *end4-1* mutation. Anti-GFP blotting of cell lysates was used to quantify the pools of full length Chs3-GFP and free GFP. The number below each lane indicates the mean (+s.e.m.) proportion of free GFP to the total GFP signal in the lane calculated from a minimum of three independent measurements. Statistical significance is indicated. Anti-Pgk1 blotting was used to control for loading. The positions of molecular weight (kDa) of protein standards are indicated on the left side of the gel. (**C**) Vacuolar localization of Mup1-GFP requires endocytosis. Micrographs showing Mup1-GFP in *mih1-1* (*chs6Δapl2Δmih1-1*) and *mih1-1end4-1* (*chs6Δapl2Δmih1-1end4-1*) cells cultured in medium lacking methionine are shown. A representative medial optical plane of a z-series is shown. Scale bar, 5 μm. (**D**) Cleavage of GFP from Mup1-GFP is attenuated by *end4-1* mutation. The data are presented as described in the legend to panel B. Due to the small amount of free GFP, the difference is not statistically significant (n.s.).

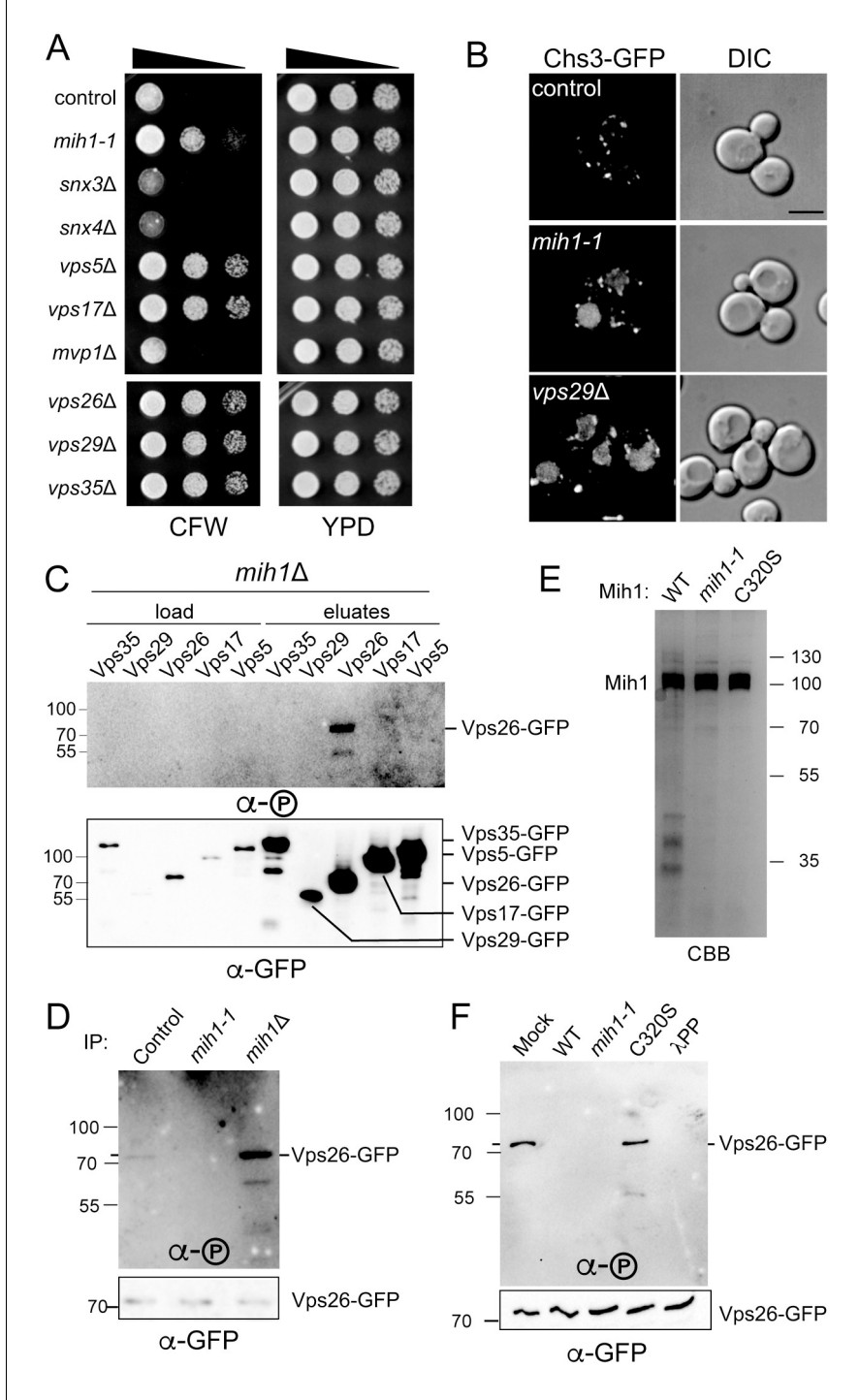

**Figure 4.** Mih1 regulates the phosphorylation state of Vps26 retromer subunit. (**A**) Growth of strains lacking genes encoding endosomal sorting nexins and retromer deletions in the presence of CFW. Serial dilutions of control (*chs6Δapl2Δ*), *mih1-1* (*chs6Δapl2Δmih1-1*), *snx3Δ* (*chs6Δapl2Δsnx3Δ*), *snx4Δ* (*chs6Δapl2Δsnx4Δ*), *vps5Δ* (*chs6Δapl2Δvps5Δ*), *vps17Δ* (*chs6Δapl2Δvps17Δ*), *mvp1Δ* (*chs6Δapl2Δmvp1Δ*), *vps26Δ* (*chs6Δapl2Δvps26Δ*), *vps29Δ* (*chs6Δapl2Δvps29Δ*) and *vps35Δ* (*chs6Δapl2Δvps35Δ*) cells were spotted onto rich medium with or without 50 μg/ml CFW for three days at 30°C. (**B**) Localization of Chs3-GFP in control (*chs6Δapl2Δ*), *mih1-1* (*chs6Δapl2Δmih1-1*) and *vps29Δ* (*chs6Δapl2Δvps29Δ*) (as a representative retromer deletion). A representative medial optical plane of a z-series is shown. Scale bar, 5 μm. (**C**) Endogenously expressed Vps35-GFP, Vps29-GFP, Vps26-GFP, Vps17-GFP or Vps5-GFP fusion proteins were immunopurified from *chs6Δapl2Δmih1Δ* cells and samples of the collected material

*Figure 4 continued on next page*

*Figure 4 continued*

were analyzed by immunoblotting with a pan phosphoryl amino acid antiserum (top) or anti-GFP (bottom). An amount of cell lysate equivalent to 0.5% of the starting material for the purifications was run in the 'load' lanes. The positions of molecular weight (kDa) protein standards are indicated on the left side of the gels. (D) Vps26 phosphorylation status. Endogenously expressed Vps26-GFP fusion protein was immunopurified from *chs6Δapl2Δ* (control), *chs6Δapl2Δmih1-1* (*mih1-1*) and *chs6Δapl2Δmih1Δ* (*mih1Δ*) cells and subjected to immunoblot assay with a pan phosphoryl amino acid antiserum (top) or anti-GFP (bottom). The molecular weight (kDa) of protein standards is indicated on the left side of the gels. (E) Purified Mih1 proteins. The indicated Mih1 proteins were purified from yeast cell lysates. A coomassie blue stained gel of aliquots of the eluted fractions used for activity assays is shown. (F) In vitro assay of Vps26 dephosphorylation by Mih1. Purified Mih1 proteins or lambda protein phosphatase were incubated with immunopurified Vps26-GFP. Equivalent portions of each reaction were examined by immunoblot with a pan phosphoryl amino acid antiserum (top) or anti-GFP (bottom). 'Mock' indicates a reaction that received no purified phosphatase.

type and Mih1-1 phosphatases, but not Mih1(C320S), dephosphorylate Vps26, which was confirmed by comparing to the products of a reaction using pure protein phosphatase derived from bacteriophage lambda (*Figure 4F*). Accumulation of phosphorylated Vps26 in *mih1Δ* cells, decreased phosphorylation of Vps26-GFP in *mih1-1* cells, and the demonstration that phosphorylated Vps26 is a substrate for purified Mih1 in vitro, lead us to conclude that Vps26 is a *bona fide* substrate of Mih1 in vivo.

Ten phosphorylated residues in Vps26 have been identified (*Stark et al., 2010*) (*Bodenmiller et al., 2008*), notably including five serine residues located within a loop that connects two β strands (corresponding to 'loop 6' in the structure of mammalian Vps26), as well as several positions corresponding to structurally significant features of arrestin family proteins, to which Vps26 is structurally related. To address the significance of Vps26 phosphorylation status controlled by Mih1, we first compared localizations and steady state abundances of GFP- and/or FLAG epitope-tagged Vps26 and Vps35 in wild-type, *mih1-1*, and *mih1Δ* cells; no differences were noted between these strains, suggesting that Mih1 does not control organelle targeting or turnover of retromer proteins (not shown). Therefore, as an indicator of retromer function, we assessed CFW resistance of mutant strains in which annotated phosphorylated residues in Vps26 were changed to alanine (Ala), and observed that no single point mutation affected CFW resistance of the cells (not shown). We therefore removed most of loop 6 (22 of 33 amino acids) including the five annotated phosphorylated serine and threonine residues, using the structure of human Vps26A (*Shi et al., 2006*) as a guide to avoid distorting the global fold of the mutant protein. Deletion of loop 6 ('Vps26ΔL6') confers increased sensitivity to CFW (*Figure 5A*), suggesting that this enhances plasma membrane localization of Chs3-GFP. That is, deletion of this loop appears to potentiate retromer-mediated recycling of Chs3. Consistent with residues of loop 6 being targets of Mih1 activity, the amount of phosphorylated Vps26ΔL6 detected by immunoblotting was reduced by ~50% compared to native Vps26 (*Figure 5B*). Next, each of the known and potential sites of phosphorylation of loop 6 were substituted with a phosphomimetic or non-phosphoryl residues and growth on CFW medium was scored (*Figure 5C*). Similar to cells with the loop 6 deletion, substitution of all serine (Ser) and threonine (Thr) residues with glutamate (Glu) results in enhanced CFW sensitivity (*Figure 5C*, rows 1, 5), suggesting that phosphorylation of these residues potentiates retromer-mediated recycling. Importantly, these substitutions exerted this effect in the context of the *mih1-1* mutation (*Figure 5C*, compare rows 2 and 3), correlating the charge on these residues with the CFW[R] phenotype of *mih1-1* cells. In contrast, substitution of these residues with non-phosphoryl alanine residues conferred modest resistance CFW (*Figure 5C*, compare rows 1 and 8), indicating that these substitutions diminish retromer function. Thus, the data argue that Mih1 regulates retromer-dependent recycling in part by controlling the phosphorylation status of Vps26 loop 6 residues. We note that the data also suggest Mih1 has additional targets that affect the growth response to CFW, as deletion of *mih1* is epistatic to the Vps26 loop 6 (A) substitutions (*Figure 5C*, compare rows 7 and 8).

Retromer contributes to Chs3 recycling in wild-type cells (*Arcones et al., 2016*), and especially in *chs6Δapl2Δ* cells (*Figure 4A*). Thus, Chs3 should possess a recycling signal that is recognized by retromer. As previously noted (*Weiskoff and Fromme, 2014*), a 'YYL' sequence (amino acids 12–14)

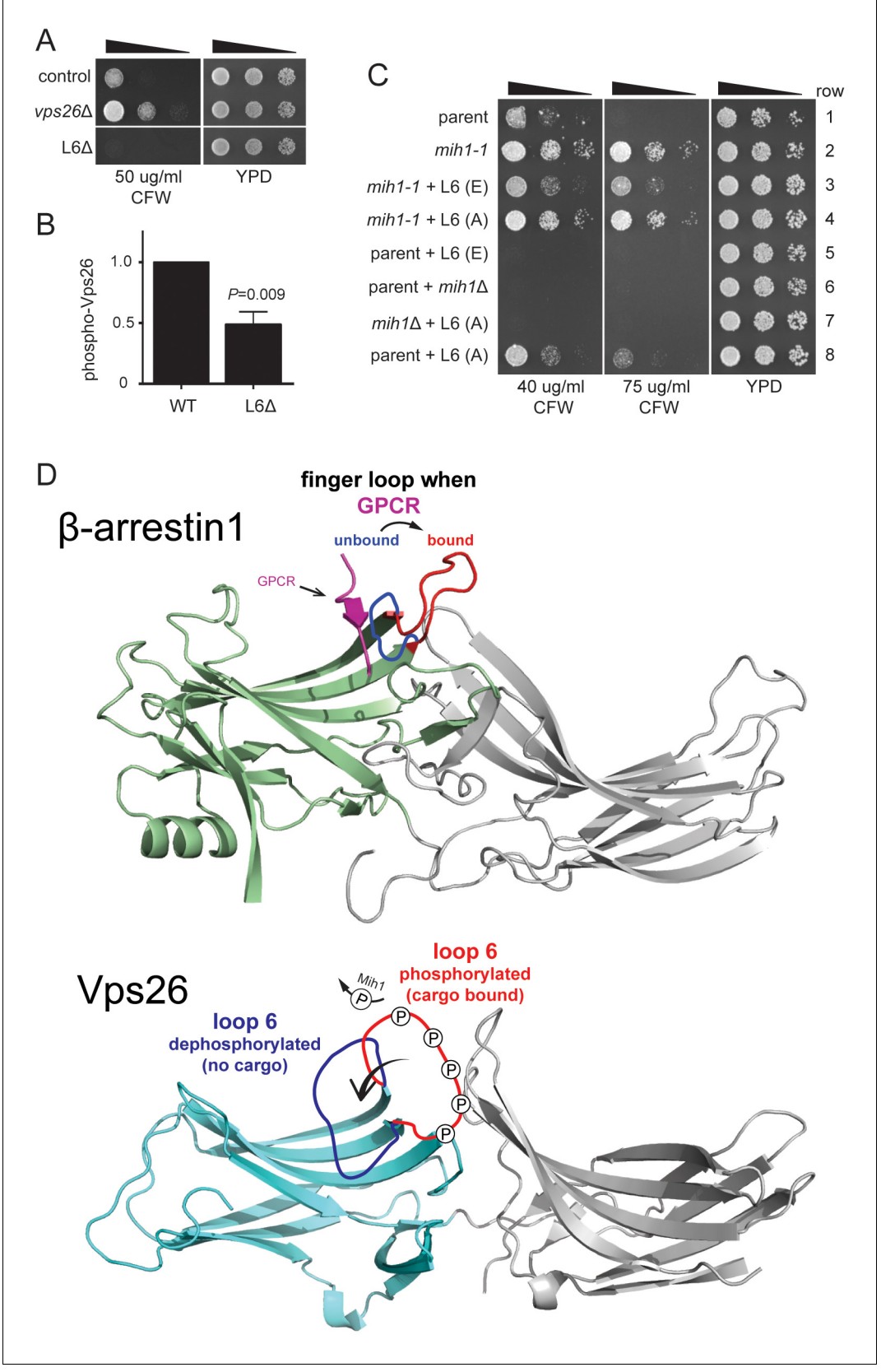

**Figure 5.** Substitutions within Vps26 loop 6 influence cargo recycling. (**A**) Deletion of loop 6 of Vps26 confers CFW sensitive growth. Serial dilutions of control (*chs6Δapl2Δ*), *vps26Δ* (*chs6Δapl2Δvps26Δ*) and L6Δ
*Figure 5 continued on next page*

*Figure 5 continued*

(*chs6Δapl2Δvps26ΔL6*) cells were spotted on YPD medium with or without CFW at the indicated concentration for three days at 30°C. (B) Vps26 loop 6 is de-phosphorylated by Mih1 in vivo. The amount of phosphorylated Vps26ΔL6 -GFP relative to native Vps26-GFP is shown. The mean (+s.e.m.) of three independent determinations is plotted. Statistical significance is indicated. (C) Phospho-mimetic substitutions in loop 6 of Vps26 confer sensitivity to CFW. Serial dilutions of control (*chs6Δapl2Δ*), mih1-1 (*chs6Δapl2Δmih1-1*), mih1−1 + L6 (E) (*chs6Δapl2Δmih1-1vps26*-loop 6 (E)), mih1−1 + L6 (A) (*chs6Δapl2Δmih1-1vps26*-loop 6 (A)), L6 (E) (*chs6Δapl2Δvps26*-loop 6 (E)), mih1Δ (*chs6Δapl2Δmih1Δ*), mih1Δ + L6 (A) (*chs6Δapl2Δmih1Δvps26*-loop 6 (A)) and L6 (A) (*chs6Δapl2Δvps26*-loop 6 (A)) cells were spotted on YPD medium with or without CFW at the indicated concentration for three days at 30°C. (D) Top: Structure of human β-arrrestin1 bound to a peptide derived from the sequence of the V2 vasopressin receptor (*Shukla et al., 2013*) (PDB: 4JQI). The 'finger loop', which corresponds to loop 6 of Vps26, is shown in the inactive (i.e., GPCR unbound) and active (GPCR bound) conformations to illustrate the manner in which it occludes GPCR binding in the inactive state. Bottom: Structure of human Vps26a (*Shi et al., 2006*) (PDB: 2FAU) with the extended loop 6 of yeast modeled in the dephosphorylated and phosphorylated conformations. We suggest that phosphorylation relieves an auto-inhibited conformation of the loop that antagonizes cargo recognition and that dephosphorylation by Mih1 restores the auto-inhibited state.

located within the N-terminal cytoplasmic segment of Chs3 is a candidate and the results of three experiments confirmed this. First, substitution of these three residues with Ala results in CFW resistance (*Figure 6A*). Second, an immobilized fusion protein composed of the first 52 amino acids of Chs3 fused to glutathione S transferase (GST) captures both Vps26 and Vps35 retromer subunits from cell lysates (*Figure 6B*). We note that a previous report found that isolated human Vps26 recognizes the retromer sorting motif of the SorLA protein (*Fjorback et al., 2012*), and that in our experiments Vps26 was slightly, but reproducibly enriched on this matrix relative to Vps35. As Vps26 is the least tightly associated yeast retromer subunit (*Seaman et al., 1998*), these data suggest that Vps26 may directly recognize GST-Chs3(1-52) while Vps35 capture is indirect. Third, ligand-binding assays employing pure recombinant Vps26 and GST-Chs3(1-52) shows that Vps26 does indeed recognize GST-Chs3(1-52) (*Figure 6C*), while there is no detectable binding to GST with an irrelevant C-terminal sequence (*Figure 6C*). Binding of Vps26 to GST-Chs3(1-52) is saturable, with an estimated equilibrium dissociation constant, $K_D$, of 19 µM (*Figure 6D*). In contrast, binding of Vps26 to a mutant GST-Chs3 fusion protein in which the YYL sequence was changed to AAA was not saturated, even at 100 µM Vps26 ($K_D$ >40 µM). These results confirm that Vps26 recognizes this sequence and, in light of the data demonstrating retromer-dependent trafficking of Chs3, indicate that this sequence constitutes a retromer-dependent sorting signal.

As the results of the cell-based experiments suggest that loop 6 impedes retromer-dependent recycling of Chs3, we measured the affinity of Vps26 lacking this loop (Vps26ΔL6) to GST-Chs3(1-52). This form of Vps26 exhibits a nearly 4-fold higher affinity for GST-Chs3(1-52) ($K_D$ of 5.2 µM) (*Figure 6D*). In close agreement with this measurement, the loop 6 phosphomimetic form of Vps26 (Vps26(L6E) binds GST-Chs3(1-52) with a $K_D$ of 4.3 µM (*Figure 6D*). Taken together, the data correlate an increase in the affinities with which the Chs3 recycling signal is recognized by Vps26 with sorting outcome; an increase in the affinity of cargo resulting from phosphorylation of loop 6 results in enhanced recycling of Chs3.

How does phosphorylation of Vps26 control the affinity with which Vps26 recognizes cargo? A role for Vps26 in cargo recognition had been raised previously on the basis of its structural homology to arrestin proteins (*Collins et al., 2008*; *Shi et al., 2006*), which serve as cargo adapters for clathrin-mediated endocytosis of GPCRs (*Kang et al., 2014*; *Moore et al., 2007*), and the finding that Vps26 binds the retromer sorting motif of SorLA (*Fjorback et al., 2012*). Atomic structures of activated β-arrestin1 (*Kim et al., 2013*), especially that of β-arrestin1 bound to a phosphorylated peptide derived from the human V2 vasopressin receptor (termed 'V2Rpp') (*Shukla et al., 2013*) (*Figure 5D*), potentially provide remarkable insight into this question. Arrestins are composed of two tandem β sandwich domains whose relative orientation toggles between an inactive conformation and an active conformation that binds ligand-activated, phosphorylated GPCR (*Kim et al., 2013*; *Shukla et al., 2013*, *2014*). The phosphorylated C-terminal segment of GPCR binds to the N-terminal domain in a concave 'cup' that is partly occluded in the inactive conformation by a loop, termed the 'finger loop,' that connects two strands within a β sheet (*Figure 5D*) (*Shukla et al.,*

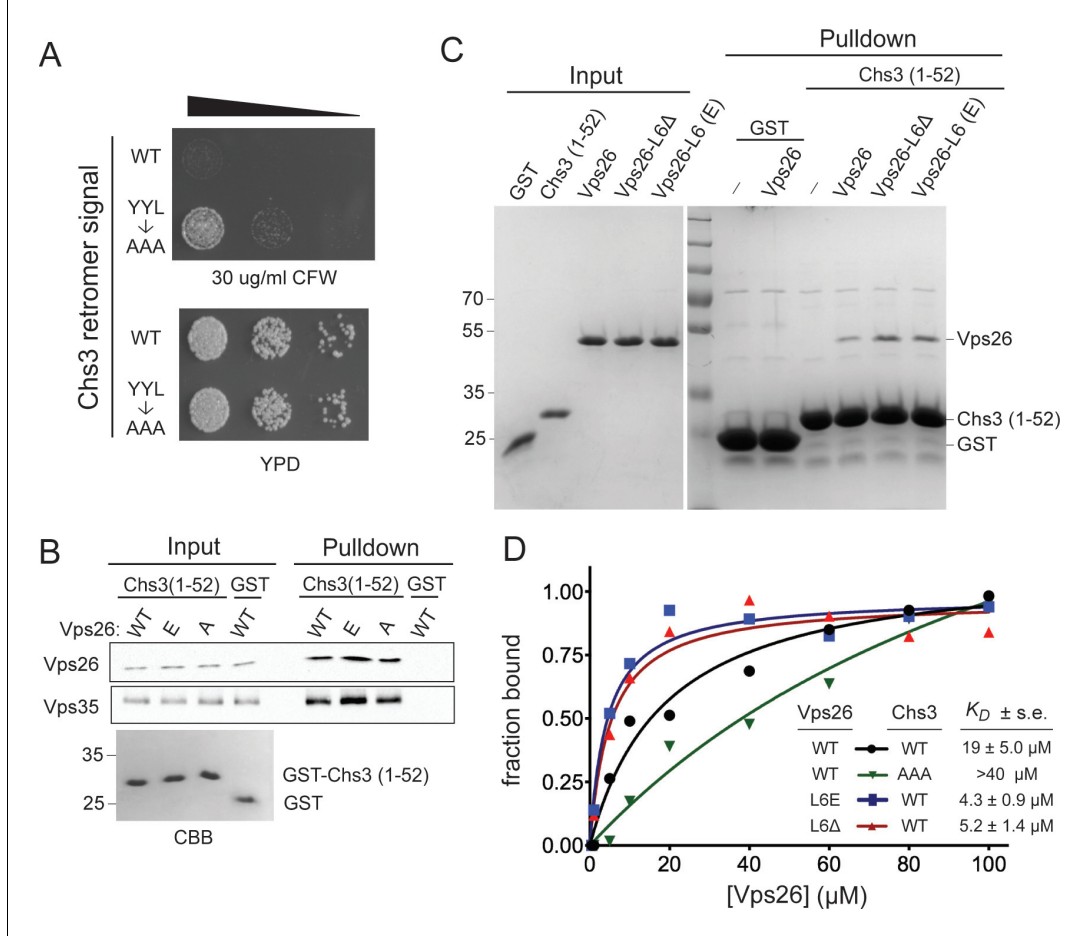

**Figure 6.** Vps26 loop 6 controls affinity for a retromer sorting signal. (A) Substitution of a putative retromer sorting motif in Chs3 confers resistance CFW. Chs3 residues $Y_{12}YL$ were changed to alanine in *chs6Δapl2Δ* cells. Serial dilutions of control (*chs6Δapl2Δ*) and mutant cells were spotted on calcofluor plates (30 µg/ml) and grown for three days at 30°C. (B) The Vps26 and Vps35 retromer subunits are captured by beads presenting Chs3 residues 1–52. Lysates from cells expressing GFP tagged Vps26 (or its variants) or Vps35-FLAG were incubated with immobilized GST-Chs3 (amino acids 1–52; 'GST-1–52') or GST. An amount of cell lysate equivalent to 0.5% of the starting material for the purifications was run in the 'load' lanes. The relative enrichment of each protein in the pulldown fraction (means and standard deviations of three independent experiments) is indicated. A coomassie blue-stained gel of 3% of each starting affinity resins is shown below the immunoblots. The molecular weight (kDa) of protein standards is indicated on the left side of the gels. (C) Vps26 binds directly to the YYL Chs3 sorting signal. Binding assays of pure, 6xHis-tagged wild-type Vps26, Vps26 loop 6 deletion (Vps26ΔL6) and loop 6 phosphomimetic Vps26 (Vps26-L6(E) to GST-Chs3(1-52) or the same portion of Chs3 with alanines substituted for the YYL sequence (GST-Chs3(YYL->AAA)) are shown. The input fractions show one microgram of the starting material used for each assay. The positions of molecular weight (kDa) protein standards are indicated on the left side of the gel. (D) Loop 6 controls affinity of Vps26 for a retromer sorting signal. Equilibrium binding measurements of pure, recombinant wild-type Vps26, Vps26L6Δ, and Vps26-L6(E) for the GST-Chs3(1-52) fusion protein, and wild-type Vps26 for the GST-Chs3(YYL->AAA) fusion protein, are shown. The points plotted in the graphs represent the mean fraction of bound Vps26 proteins, calculated from single measurements using three independent protein preparations of each Vps26 protein. The calculated $K_D$ values (±s.e.) are listed. Saturation binding to the GST-Chs3(YYL->AAA) fusion protein was not observed, so a dissociation constant could not be calculated.

2013). In the active, receptor-bound conformation of β-arrestin1, the finger loop extends from the domain, allowing it to contact the membrane core of the GPCR and exposing the binding site of the GPCR C-terminal segment (*Figure 5D*) (*Shukla et al., 2013*; *Szczepek et al., 2014*).

The finger loop of β-arrestin1 corresponds to the phosphorylated loop 6 of Vps26 (*Figure 5D*). It is unlikely that loop 6 directly recognizes the retromer sorting signal. Rather, we suggest that it functions as a gate that controls access of the β sheet of N-terminal domain to cargo (*Figure 5D*). In its unphosphorylated state, the extended loop would antagonize cargo engagement via an auto-inhibited conformation, akin to the conformation of the finger loop of β-arrestin1 in the inactive state,

and phosphorylation of loop residues would displace it from occluding access to the surface of the $\beta$ sheet, akin to the displacement of the $\beta$-arrestin-1 finger loop in the active conformation. This mechanism provides for a tunable system for adjusting the abundance of proteins on the cell surface via recycling regulated by kinase-phosphatase networks that are responsive to metabolic and environmental cues. Surprisingly, neither the *mih1-1* or *mih1Δ* mutations affect the distribution of Vps10, a well characterized retromer cargo that cycles between the Golgi apparatus and endosome (*Figure 2—figure supplement 1*), indicating the existence of at least one additional mode of cargo recognition by retromer, speculated to be mediated by the Vps35 retromer subunit (*Nothwehr et al., 1999*, *2000*). Consistent with this, a recently published structure of a Vps35-Vps26-Snx3 complex bound to a peptide containing a retromer sorting signal revealed an additional mode of cargo recognition involving the Snx3-Vps26 interface (*Lucas et al., 2016*). Multiple modes of cargo recognition by retromer provide a mechanistic explanation for the diversity of cargo-specific sorting factors that function with retromer, such as the Snx3, SNX-BAR, and Snx27 sorting nexins (*Cullen, 2008*; *Cullen and Korswagen, 2012*; *Strochlic et al., 2007*). As cargo recognition plays a key role in targeting soluble retromer to the endosome membrane (*Harrison et al., 2014*; *Lucas et al., 2016*), an important implication of our findings is that distinct cargo recognition modes could drive the formation of distinct retromer-containing complexes on the endosome membrane.

A future challenge is to identify the signaling pathways and regulatory inputs that modulate plasma membrane composition via retromer-mediated recycling. Importantly, the identification of Mih1 as a physiological regulator of retromer function reveals a new facet of function for CDC25 family phosphatases, which are currently thought to act solely on cyclin-dependent kinases to control cell cycle progression (*Boutros et al., 2007*). The activities of CDC25 family phosphatases are subject to intricate regulation via kinases and phosphatases that respond to a variety of cellular stimuli and impinge on various cell cycle checkpoints. One possible regulator of Mih1 function with regard to Chs3 recycling is PKC, which is has been shown to control Mih1 signaling (*Yano et al., 2013*) and to promote Chs3 trafficking from the TGN to the plasma membrane via the exomer pathway (*Valdivia and Schekman, 2003*). As PKC is a core component of the cell wall integrity pathway, Mih1-mediated regulation of Chs3 recycling by retromer might reflect a stress response to defects in the cell wall.

The results presented here indicate that, in addition to the role that yeast Mih1 plays in cell cycle progression, it also controls trafficking of plasma membrane proteins that impact nutrient homeostasis and cell wall remodeling – key axes of cell growth control. Over-expression of CDC25 proteins is observed in some human cancers, leading to speculation that gain-of-function dysregulation, akin to the Mih1-1 protein characterized in this study, contributes to cell transformation. However, it has not been possible to attribute the effects of increased CDC25 expression levels solely to inappropriate progression through the cell cycle and our study raises the possibility that this may be due, in part, to unappreciated activities of CDC25 proteins on non-CDK substrates.

## Materials and methods

### Yeast strains and culture conditions

All yeast strains were constructed in BY4742 (*MATα his3-1*, *leu2-0*, *met15-0*, and *ura3-0*) (Open Biosystems/Thermo Scientific, Waltham, MA) by homologous recombination of gene-targeted, polymerase chain reaction (PCR)-generated DNAs using the method of Longtine et al (*Longtine et al., 1998*). Mutant strains were derived either from the EUROSCARF *KANMX* deletion collection (Open Biosystems/Thermo Scientific, Waltham, MA) or by replacement of the complete reading frame with a *HIS3MX6*, *NATMX6* or *URA3* cassette. The introduction of point or truncation mutations into the genome was performed according to the established methods (*Gray et al., 2004*, *2005*; *Toulmay and Schneiter, 2006*). To make the Vps26-E mutant, all serine and threonine codons in loop 6 region of Vps26 (87–113 aa) were replaced by glutamate codons; To make the Vps26-A mutant, all serine and threonine codons in loop 6 region of Vps26 were replaced by alanine codons. To express TAP-tagged Mih1, a TAP tag DNA construct was integrated immediately upstream of the native stop codon of strain TVY614 (*MATα, ura3–52, his3-Δ200, trp1-Δ 901, lys2–801, suc2-Δ9, leu2–3, 11, pep4Δ::LEU2, prb1Δ::HISG, prc1Δ::HIS3*). Next, DNAs generated by PCR using pFA-kanMX6-PGAL1 and pBS1539 as templates were used to integrate the *GAL1* promoter immediately

upstream of the native *MIH1* start codon and a TAP tag immediately upstream of the native stop codon, respectively. To construct similar strains expressing the *mih1-1* or C320S alleles, the *MIH1* locus of TVY614 was first replaced with the *URA3* gene. These strains were then transformed with DNA generated by PCR of the *mih1* locus using genomic DNA from strains containing the *mih1-1* or C320S alleles as template. Transformants were selected on plates containing 5-fluoroorotic acid and replacement of the *URA3* locus by the *mih1* allele was confirmed by DNA sequencing. Next, a TAP tag and the *GAL1* promoter were integrated as described for the native *MIH1* locus. All newly constructed strains were confirmed to present relevant autotrophic/drug resistance markers (indicating that they were derived from the BY4742 parent) and chromosomal manipulations were confirmed by PCR amplification and DNA sequencing of the relevant locus.

Cells were grown in rich YP medium (2% bacto-peptone, 1% yeast extract, 2% glucose), or complete synthetic medium (Sunrise Science Product, San Diego, CA). Corresponding nutrient(s) were omitted from media to maintain selection for auxotrophic markers and/or plasmids (*Sherman et al., 1979*). Cells used to produce TAP-tagged Mih1 were grown at 23°C in YP +2% raffinose+0.1% glucose until an $OD_{600} \approx 0.6$, when galactose was added to 2% and growth continued overnight. To obtain calcofluor-resistant mutants, *chs6Δapl2Δ* cells were spread onto YPD plates containing 100 µg/mL CFW, and the plates were incubated at 30°C for three days. Colonies were picked and restreaked on a CFW plate. To identify mutant strains with CFW$^R$-conferring mutations in expected genes, each mutant strain was mated to *chs3Δ, chs4Δ, chs5Δ, chs7Δ and pfa4Δ* strains (in the BY4741 background) and CFW$^R$ was evaluated. If the resulting diploid cells were CFW$^R$, the candidate was determined to have a recessive mutation in the gene which is lacking in the mating partner and the strain was discarded. Candidates that were not complemented by any 'tester' strain were retained. Whole genome DNA sequencing of the parent and *mih1-1* strains was used to identify the lesion in the *mih1-1* strain. DNA sequencing was performed by the Iowa Institute of Human Genetics, University of Iowa using the Illumina HiSeq platform. Genome assembly was done using Lasergene software (DNASTAR, Madison, WI).

For spot growth assays, cells from cultures were grown to $OD_{600} \approx 0.5$ in liquid YPD medium were washed with water and fresh YPD medium, and then a 10-fold dilution series was spotted onto plates with a series of concentrations of CFW (0–150 µg/mL). The plates were assessed after 2–3 days of incubation at 30°C.

## Immunoblot and immunoprecipitation

Cells from cultures were grown to $OD_{600} \approx 0.5$ in liquid complete synthetic medium or YPD medium. Equal amounts of cells were collected and TCA-precipitated, and then proteins were resolved by SDS-PAGE. Primary antibodies in these studies include: anti-GFP (1:2000; Roche), anti-Pgk1 (3-phosphoglcerate kinase) (1:10,000; Invitrogen), anti-phosphoserine/threonine/tyrosine antibody (1:500; Abcam), anti-FLAG (1:5000; sigma), anti-His (1:1000; Biolegend) and anti-Phospho-cdc2 (Tyr15) (1:1000; Cell Signaling Technology).

For immunoprecipitation of GFP tagged retromer subunits, cells were grown to $OD_{600} \approx 0.5$ in complete synthetic medium, and then were washed and frozen in liquid nitrogen prior to disruption by bead beater in lysis buffer (20 mM HEPES pH = 7.4, 150 mM KOAc, 5% glycerol, 1% $N_{P40}$, proteinase inhibitor cocktail (Roche), PhosSTOP (Roche)). GFP-nAb magnetic beads (Allele Biotech) were used to capture GFP tagged proteins. After binding, the beads were extensively washed and then eluted with SDS-PAGE sample buffer.

## Light microscopy and image analysis

Cells were grown to $OD_{600} \approx 0.5$ in liquid medium were mounted in growth medium on a microscope slide. Image stacks were collected at 0.3-µm z increments on a DeltaVision workstation (Applied Precision) based on an inverted microscope (IX-70; Olympus) using a 100 × 1.4 NA oil immersion lens. Images were captured at 24°C with a front illuminated sCMOS, 2560 × 2160 pixels camera and deconvolved using the iterative-constrained algorithm and the measured point spread function. Image analysis and preparation were done using Softworx 6.1 (Applied Precision Instruments) and ImageJ v1.50d (*Rasband, 2015*).

## Protein expression and purification

Plasmids, pET28a-Vps26 and pGST-Chs3 (1-52 amino acid) (*Weiskoff and Fromme, 2014*), were used to express 6XHis-Vps26 and GST-Chs3 (1-52) fusion proteins in *Escherichia coli*. Cells were grown in LB media containing the appropriate antibiotics in baffled flasks filled to 20% of the total volume to $OD_{600} \approx 0.5$–0.7 and IPTG was added to a final concentration of 0.5 mM and growth was continued for 3 hr at 37°C. Proteins were purified in batch using GSH beads (GE Healthcare Life Sciences) or Ni-NTA Agarose (Qiagen) and confirmed by SDS-PAGE and immunoblot with corresponding antibodies against each tag. Purified protein concentration was quantified by BCA assay (Pierce).

Frozen Mih1-TAP cells were thawed and then lysed (50 mM Tris, pH 7.4, 300 mM $NaCl_2$, 1 mM DTT, proteinase inhibitor (Roche), 0.5%Triton) using an EmulsiFlex-C3 system (Avestin). Lysates were clarified by ultracentrifugation at 100,000 x *g* for 1 hr at 4°C. IgG agarose beads (GE Healthcare) were rinsed three times in lysis buffer. Beads were added to cleared lysate and incubated 4 hr at 4°C with rotation. Beads were washed (10 × 1 ml) and suspended in tobacco etch virus (TEV) protease buffer (50 mM Tris, pH 7.4, 150 mM $NaCl_2$, 0.5 mM EDTA, 1 mM DTT). TEV protease was added to the buffer and incubated overnight at 4°C. To reduce the association of Mih1-TAP with IgG beads, IgG beads were precipitated and rinsed in calmodulin binding buffer (20 mM Tris, pH 7.4, 300 mM $NaCl_2$, 4 mM $CaCl_2$, 1 mM magnesium acetate, 1 mM imidazole, 10 mM 2-mercaptoethonal). Calmodulin magnetic beads were rinsed, suspended in calmodulin binding buffer, and added directly to precipitated IgG beads. The mixed beads were incubated at 4°C with rotation for 4 hr. Calmodulin magnetic beads were isolated magnetically and washed extensively. Mih1-TAP proteins were eluted with elution buffer (20 mM Tris, pH 8.0, 150 mM $NaCl_2$, 10 mM EGTA and 10 mM 2-mercaptoethanol).

## Protein phosphatase assay

Mih1, Mih1-1, and Mih1(C320S) were purified as described above. Vps26-GFP was immunoprecipitated from *chs6Δapl2Δmih1Δ* cells. The immobilized Vps26-GFP was incubated with purified Mih1 proteins in the phosphatase assay buffer (40 mM Tris-HCl, pH 8.0, 100 mM NaCl, 1 mM EDTA, 5 mM DTT) at 30°C for 1 hr. The immobilized Vps26-GFP was rinsed, boiled and subjected to immunoblotting with a pan phosphoryl amino acid antiserum.

## In vitro binding assays

Purified GST-Chs3 (1-52) or GST was immobilized to glutathione sepharose. Immobilized GST-Chs3 (1-52) or GST (1 μM) was then mixed with purified WT or mutant 6XHis-Vps26 proteins at the indicated concentration. After overnight incubation, the beads were washed (10 × 1 ml) with PBS buffer (137 mM NaCl, 2.7 mM KCl, 10 mM $Na_2HPO_4$, 1.8 mM $KH_2PO_4$, 1 mM TCEP, pH = 7.4). The bound 6XHis-Vps26 proteins were extracted from the beads with SDS-PAGE sample buffer and detected by immunoblot with an anti-His antibody (Biolegend). Immunoblot signals were quantified using Image Lab software (Biorad).

## Statistical analyses

Prism 7 software (GraphPad, La Jolla, CA) was used to calculate means, standard deviations and standard errors of mean values. The Student's unpaired *t* test was used for statistical analyses. A $p<0.05$ value was considered to indicate statistical significance.

## Acknowledgements

We thank Rob Piper (University of Iowa) for assistance with genome sequencing and analysis, Doug Kellogg (University of California, Santa Cruz) for discussions regarding Mih1, Chris Fromme (Cornell University) for the GST-Chs3(1-52) expression plasmid, and colleagues for critical comments on the manuscript. Research reported in this publication was supported by the National Institute of General Medical Sciences of the National Institutes of Health under award number GM060221.

## Additional information

### Funding

| Funder | Grant reference number | Author |
|---|---|---|
| National Institutes of Health | GM061221 | Christopher G Burd |

The funders had no role in study design, data collection and interpretation, or the decision to submit the work for publication.

### Author contributions

T-ZC, TAP, Investigation; CGB, Conceptualization, Data curation, Formal analysis, Supervision, Funding acquisition, Investigation, Writing—original draft, Project administration, Writing—review and editing

### Author ORCIDs

Tabitha A Peterson, http://orcid.org/0000-0003-0780-5795
Christopher G Burd, http://orcid.org/0000-0003-1831-8706

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
