## [Decision Letter]

Thank you for submitting your article "A CDC25 family protein phosphatase gates cargo recognition by the Vps26 Retromer subunit" for consideration by *eLife*. Your article has been reviewed by three peer reviewers, one of whom Suzanne Pfeffer, is a member of our Board of Reviewing Editors, and the evaluation has been overseen by Randy Schekman as the Senior Editor. The reviewers have discussed the reviews with one another and the Reviewing Editor has drafted this decision to help you prepare a revised submission.

Here, Cui et al. use a genetic screen to identify a mutant form of the Mih1 phosphatase that alters the recycling of specific proteins at endosomes. They show that Mih1 de-phosphorylates the retromer subunit Vps26, and suggest that phosphorylation of "loop 6" blocks an autoinhibitory interaction to enhance the binding of a subset of retromer cargo. The discovery of a new mechanism to regulate cargo recognition by the retromer complex is novel and important. However, data supporting key elements of the model could be strengthened by additional controls and quantitative analyses throughout, as described below.

Major comments:

1) Better quantification of cargo recycling defects is needed, and statistical significance should be reported. While statistical tests are described in the methods, p-values were not reported in the figures or results. Because the cargo missorting phenotypes are relatively weak, careful quantification and statistics are needed to show differences are significant. As an example, it was not obvious that *mih1∆* significantly decreased Chs3-GFP or Mup1-GFP cleavage in Figure 2 or 2D as suggested on pages 7 and 8. (Please add quantification to all figures and supplements with p values where possible. Completely independent experiments should be compared.)

For other cargo, fluorescence microscopy is the only technique used to evaluate missorting, and no quantification is provided. It would strengthen the manuscript to use additional measures of Can1 and Mup1 localization. The Mup1-pHluorin reporter developed by Wendland lab (Prosser et al. Traffic. 2010) is widely used, and canavanine resistance can detect retromer-dependent missorting of Can1, particularly in a snx41 background (see Shi et al., MBoC 2011).

2) Evidence that Mih1 dephosphorylates the Vps26 L6 loop should be bolstered. While the L6E mutant does phenocopy *mih1∆, mih1∆* is epistatic to L6A and this should not be the case if *mih1∆* phenotypes result from enhanced L6 phosphorylation. Moreover, *mih1-1* is epistatic to both L6A and L6E. To explain the epistasis, the authors suggest Mih1 has additional targets that affect growth on CFW. If this is the case, other measures of cargo missorting should be used that do not rely on CFW, such as the Mup1-pHluorin assay described above.

To test the hypothesis that Mih1 dephosphorylates the L6 loop, the phosphorylation status of the L6A, L6E and L6∆ mutants should be examined in the *mih1∆* strain, and compared to wild type Vps26 as shown in Figure 4.

3) Additional experiments are needed to establish that YYL is the retromer signal. To show that YYL is the retromer-binding site, a YYL->AAA mutant form of the GST-Chs3(1-52) construct should be included in one or more of the retromer-binding assays shown in Figure 6. Other experiments to strengthen this conclusion could include the demonstration of an equivalent degree of suppression by YYL->AAA, *vps26∆*, and the YYL->AAA *vps26∆* double mutant (note the CFW concentration is not indicated in 6A, making it difficult to compare with other figures). Fluorescence microscopy could also be used to test if Chs3-YYL-GFP mutant is localized to the vacuole, similar to Chs3-GFP in a retromer mutant.

4) CFW data suggest removing the L6 loop makes Vps26 work *better*. This is actually pretty cool, so it would be nice if the authors had another line of evidence to support this result.

5) In Figure 6, the error bars are huge and no error estimates are given, presumably because the experiments have only been done once and to quote K_D_s to 3 significant figures is inappropriate. Could ITC or thermofluor be used here? A better estimate of K_D_s would be really valuable and support the model proposed. It would also be good if the authors could show that mutations do not affect Vps35 binding but this would be dependent on being able to make recombinant Vps35. It would also be nice if a poly phosphorylated version could be used but if the kinase or a fraction containing the kinase is unknown this is probably too much to ask for and using the poly Asp/Glu mutant would suffice.

[Editors' note: further revisions were requested prior to acceptance, as described below.]

Thank you for resubmitting your work entitled "A CDC25 family protein phosphatase gates cargo recognition by the Vps26 retromer subunit" for further consideration at *eLife*. Your revised article has been favorably evaluated by Randy Schekman, Senior editor, and Suzanne Pfeffer, Reviewing editor. We appreciate the thoughtful manner in which you responded to the reviewer comments.

The manuscript has been improved significantly, and we would like to be able to present this story in *eLife* after the following issue is corrected. We believe there is an error in the estimation of the binding affinity for the AAA mutant in Figure 6; the data shown would imply only that the affinity is >40µM, not 130. Please investigate this and correct the figure accordingly. (Also, the legend indicates average of 3 preps, each done in singlicate? Duplicate? Is the error between preps or between individual samples?)

---

## [Author Response]

*1) Better quantification of cargo recycling defects is needed, and statistical significance should be reported. While statistical tests are described in the methods, p-values were not reported in the figures or results. Because the cargo missorting phenotypes are relatively weak, careful quantification and statistics are needed to show differences are significant. As an example, it was not obvious that mih1∆ significantly decreased Chs3-GFP or Mup1-GFP cleavage in Figure 2 or 2D as suggested on pages 7 and 8. (Please add quantification to all figures and supplements with p values where possible. Completely independent experiments should be compared.)*

*For other cargo, fluorescence microscopy is the only technique used to evaluate missorting, and no quantification is provided. It would strengthen the manuscript to use additional measures of Can1 and Mup1 localization. The Mup1-pHluorin reporter developed by Wendland lab (Prosser et al. Traffic. 2010) is widely used, and canavanine resistance can detect retromer-dependent missorting of Can1, particularly in a snx41 background (see Shi et al., MBoC 2011).*

We quantified cargo recycling defects by determining the proportion of full length GFP-tagged fusion protein that is processed to liberate GFP, a widely-used measure of delivery to the vacuole, i.e., failure to recycle. We report the means of three independent determinations and provide statistical analyses of significance in the revised figure legends and figures. The recycling defects observed for Chs3-GFP and Can1-GFP (Figure 2) in *mih1-1* cells are significant (*P*<0.05), or highly significant (*P*<0.05). For *mih1Δ* cells, where we observed reduced processing of the fusion proteins (probably due to a difference in the rate of endocytosis), only the Chs3-GFP is significant (*P*<0.05). Nonetheless, the trend of the Mup1 and Can1 data are highly reproducible and therefore physiologically significant. The epistasis data presented in Figure 3 was analyzed in the same manner as in Figure 2. The difference in Chs3-GFP processing in *mih1-1* and *mih1-1 end4-1* cells is significant (*P*<0.05), but the difference for the Mup1 data is not because the amount of Mup1 that is processed in *mih1-1* cells is small.

The reviews pointed out that for the GFP-tagged cargoes that are not affected by the *mih1-1* mutation (presented in the Figure 1—figure supplement 1), we relied solely on the localization patterns of the proteins in the original submission. In the revised manuscript, we more rigorously address this by quantifying the amounts of full-length fusion proteins because essentially no free GFP was observed to be liberated from any of them. We observed no statistically significant differences in protein abundances in *mih1-1* or *mih1Δ* cells.

*2) Evidence that Mih1 dephosphorylates the Vps26 L6 loop should be bolstered. While the L6E mutant does phenocopy mih1∆, mih1∆ is epistatic to L6A and this should not be the case if mih1∆ phenotypes result from enhanced L6 phosphorylation. Moreover, mih1-1 is epistatic to both L6A and L6E. To explain the epistasis, the authors suggest Mih1 has additional targets that affect growth on CFW. If this is the case, other measures of cargo missorting should be used that do not rely on CFW, such as the Mup1-pHluorin assay described above.*

*To test the hypothesis that Mih1 dephosphorylates the L6 loop, the phosphorylation status of the L6A, L6E and L6∆ mutants should be examined in the mih1∆ strain, and compared to wild type Vps26 as shown in Figure 4.*

We addressed this by measuring the amount of phospho-Vps26ΔL6 to phospho-Vps26 by immunoblotting of the Vps26-GFP proteins immuno-purified from *mih1Δ* cells, as suggested by the reviewers. We observe that deletion of loop 6 reduces the phospho Vps26 signal by approximately 50%, indicating that loop 6 phosphorylation sites accumulate in the absence of Mih1. The new data are included in Figure 5 (panel B).

3) Additional experiments are needed to establish that YYL is the retromer signal.

*To show that YYL is the retromer-binding site, a YYL->AAA mutant form of the GST-Chs3(1-52) construct should be included in one or more of the retromer-binding assays shown in Figure 6. Other experiments to strengthen this conclusion could include the demonstration of an equivalent degree of suppression by YYL->AAA, vps26∆, and the YYL->AAA vps26∆ double mutant (note the CFW concentration is not indicated in 6A, making it difficult to compare with other figures). Fluorescence microscopy could also be used to test if Chs3-YYL-GFP mutant is localized to the vacuole, similar to Chs3-GFP in a retromer mutant.*

We measured the affinity of Vps26 for a mutant version of the GST-Chs3 fusion protein in which the YYL sequence was changed to AAA. We did not observe saturation binding to this protein, even with 100 μM Vps26, indicating a dissociation constant ≥130 μM. This result indicates that this sequence is necessary for recognition by Vps26 and is, therefore, consistent with it constituting a retromer-dependent sorting signal. These data are included in Figure 6.

*4) CFW data suggest removing the L6 loop makes Vps26 work *better*. This is actually pretty cool, so it would be nice if the authors had another line of evidence to support this result.*

We agree – this is a cool result. A second line of evidence in support of this interpretation was presented in panel D of Figure 6, where we showed that a version of Vps26 lacking loop 6 binds to the portion of Chs3 containing the retromer sorting signal with approx. 4-fold higher affinity than the native protein (which contains loop 6). The increase in affinity for the Chs3 sorting signal indicates that the mutant protein “works better”.

*5) In Figure 6, the error bars are huge and no error estimates are given, presumably because the experiments have only been done once and to quote K_D_s to 3 significant figures is inappropriate.*

Evidently, we did not make clear in the original manuscript that each point represents the mean (s.d. indicated) of three independent binding experiments (not one), where *three independent protein preparations* of the four different Vps26 proteins (total of 12 preps) and the GST-Chs3 fusion proteins were assayed. Although the implementation of different protein preparations likely contributes to the variability noted by the reviewer(s) (s.d. for each point is shown), and despite the substantial effort required to prepare all of the proteins, we consider this rigorous practice. In the revised manuscript, we present the results to two significant digits and indicate the standard errors of the mean *K_D_* calculations.

*Could ITC or thermofluor be used here? A better estimate of K_D_s would be really valuable and support the model proposed. It would also be good if the authors could show that mutations do not affect Vps35 binding but this would be dependent on being able to make recombinant Vps35. It would also be nice if a poly phosphorylated version could be used but if the kinase or a fraction containing the kinase is unknown this is probably too much to ask for and using the poly Asp/Glu mutant would suffice.*

Yes, other techniques could have been used to calculate affinities. In retrospect, it would have been a good idea to identify a collaborator with access to and expertise with the suggested instrumentation. We hope, however, that the reviewers will appreciate the considerable effort that has already been dedicated to these experiments and that our limited resources preclude us from simply discarding the results in hand.

[Editors' note: further revisions were requested prior to acceptance, as described below.]

*The manuscript has been improved significantly, and we would like to be able to present this story in eLife after the following issue is corrected. We believe there is an error in the estimation of the binding affinity for the AAA mutant in Figure 6; the data shown would imply only that the affinity is >40µM, not 130. Please investigate this and correct the figure accordingly. (Also, the legend indicates average of 3 preps, each done in singlicate? duplicate? Is the error between preps or between individual samples?)*

1) Figure 6. The graph has been replaced and the K_D_ value of Vps26 for the GST-Chs3 (YYL->AAA) fusion protein has been corrected.

2) The legend to Figure 6 has been modified to account for the changes to the figure and address the question raised about replicates.

3) The K_D_ value of Vps26 for the GST-Chs3(YYL->AAA) fusion protein listed in the text has been corrected.